# Gratitude and Adolescents' Mental Health and Well-Being: Effects and Gender Differences for a Positive Social Media Intervention in High Schools

**Giacomo Bono \***  , **Taylor Duffy and Erin L. Merz**

Psychology Department, California State University, Dominguez Hills, Carson, CA 90747, USA
* Correspondence: gbono@csudh.edu

**Abstract:** Gratitude interventions can provide cost-effective support for mental health to under-resourced schools. This study aims to better understand the effects of a promising intervention Bono et al. evaluated in 2020. Using a quasi-experimental design (where classes were assigned to a thanking app, gratitude curriculum, app + curriculum, or control condition), that evaluation found that the full (combined) intervention impacted students' self-reported trait gratitude, anxiety, and subjective well-being (SWB) over six weeks, compared against only the control condition. However, here, we evaluated the individual intervention components' effectiveness on students (N = 326) using multilevel modeling. As hypothesized, the full intervention impacted students' gratitude, anxiety, and SWB, compared to the control condition, but impacted SWB more than the app-only condition, suggesting that teaching gratitude science makes thanking more meaningful. Then, we examined if stress mediated these effects. Perceived stress partially mediated the relationships of gratitude with depression and SWB and fully mediated the relationship of gratitude with anxiety. Additionally, changes in perceived stress and SWB differed by gender. Finally, we qualitatively analyzed thanks exchanges during the intervention using informal content analysis and found themes of psychological safety—a critical feature neglected in other interventions. We conclude with recommendations for optimizing school gratitude interventions.

**Keywords:** gratitude; intervention; adolescence; mental health; social emotional learning

## 1. Introduction

Society presents many challenges for adolescents today. The path to adulthood is longer, more competitive, and more uncertain than ever [1]. Contemporary society is awash with more media, information, and communications than ever in history, which increasingly taxes individuals' attentional resources [2]. Many households have both parents working and face more financial and time management challenges than ever [3], and schools must address obstacles to learning and learn to support students' mental health and well-being more effectively [4]. Unfortunately, mental health concerns disproportionately affect youth worldwide, and school-based resources in many countries are insufficient to meet these needs [5]. Overall, achieving a positive and purposeful identity is more challenging than ever for adolescents today, and schools must, nonetheless, figure out how to support their academic growth and longer-term career success. Promoting gratitude in adolescents with intervention can help alleviate these issues.

### 1.1. Adolescents in Contemporary Society

Adolescence is a vital developmental period for individuals to establish positive habits for thriving into adulthood. However, there is a discouraging trend in U.S. schools that may also be relevant in other industrialized countries, given the common cultural trends described above. From 2008 to 2017, adolescents in the U.S. have become more bored across and within secondary school grades [6]. On the other hand, another national survey finds that

over 70% of students report feeling stressed out or bored at their high school most or some of the time recently [7]. Such trends are consistent with psychopathology trends in the U.S., which have been rising among high school and college students for several decades [8].

However, declines in youths' mental health and well-being have become more acute recently. Evidence from large national surveys in the U.S. shows that undergraduate students have decreased in life satisfaction and increased in anxiety and depression symptoms from 2007 to 2018 [9]. One reason for these trends is the increased use of new media among adolescents due to the rapid spread of smartphones and social media in daily life [10]. In surveying the evidence for possible mechanisms, researchers have found that heavy use of new media has: displaced in-person social interactions, disrupted in-person interactions, interfered with sleep, contributed to toxic social media environments that expose youth to incivility and unhealthy amounts of social comparison or, worse, cyberbullying, and increased the contagion or spread of self-harm behaviors [11]. The recent COVID pandemic exacerbated this trend [12]. This has long-term ramifications, as stress and mental health concerns are major obstacles to adolescents' college success [13].

Thus, the well-being and mental health of adolescents in the U.S. and throughout the world reflects a crisis that requires easy and scalable solutions. Effective gratitude interventions in schools can provide one such solution.

*1.2. Why School-Based Gratitude Interventions Are Needed*

Gratitude is valued as a virtue of strength in all the major religions of the world [14], and interventions in schools may provide a cost-effective solution for supporting adolescents' mental health and well-being at a broad scale. Supportive social networks buffer individuals from adversity and pathology and enhance health and well-being throughout life [15], and supportive social ties with peers and teachers support the wellness and functioning of students and schools [16,17]. Gratitude is uniquely suited to developing quality social connections [18,19], but recent meta-analytic research concludes that it is unclear whether school gratitude interventions are generally effective or useful for students or schools [20]. However, two recent intervention studies show that letting adolescent students share gratitude genuinely with others and having them reflect on their thanking behavior improves the impacts of interventions on their gratitude and well-being [21,22].

With this study, we hope to inspire educators and researchers to adopt best practices for school gratitude interventions and to encourage using gratitude practices as an important tool for personalizing and improving social emotional learning (SEL). Specifically, the purpose of this study is to re-examine the high school gratitude intervention that Bono et al. [22] originally conducted to evaluate the effectiveness of its individual intervention components for supporting male and female students' mental health and well-being. The original evaluation found overall significant impacts in these areas, but it only compared an optimal intervention condition (i.e., classes that used both a gratitude curriculum and a social media-like app for sharing thanks in classrooms) against a control condition (i.e., classes that did business as usual and received no intervention). The curriculum-only and app-only interventions were not evaluated against each other or against a control condition. Additionally, the intervention was implemented in 2 high schools across 21 different classrooms, but Bono et al. did not statistically account for the nested structure of the data.

Therefore, in the current research we use multilevel modeling to account for the interdependency in the data to explore if all four intervention conditions have different effects on male vs. female students' trait gratitude, depression, anxiety, and subjective well-being (SWB) in one study. Then, we sought to better understand gratitude's role in predicting these outcomes by testing mediation models of whether gratitude's relationship with perceived stress accounts for its effects on these outcomes and whether there are any gender effects on them or on the putative mediator (i.e., perceived stress). Finally, we informally content analyzed the thanks messages that students and teachers exchanged during the intervention for themes of psychological safety. This quality has not been designed

into school gratitude interventions prior to Bono et al.'s study, but it is critical for making thanking more meaningful and impactful on individuals' mental health and well-being.

### 1.3. Supporting Social Emotional Learning in High Schools with Gratitude Practices

Studies that increase knowledge about risk factors and protective factors related to anxiety and depression are practical because they help promote well-being, which is especially needed for adolescents today. Indeed, this is a central focus of the new field of positive education and of social emotional learning (SEL) programs in schools, that are producing research and resources on the science of student and school well-being [8,23]. SEL programs significantly improve social and emotional skills, attitudes, behavior, and academic success [24,25], but they could better support moral behavior with more focus on developing caring relationships and fostering a strong school community [26]. As suggested above, including gratitude practices in SEL may help fill this shortcoming in SEL.

Developmentally, adolescents face many demands at once. They must develop a coherent set of values and beliefs and internalize norms and practices to develop their identity, socially and academically [27]. Thus, it is critical for schools to effectively support both psychosocial and academic development in students at once, to prevent personal issues from becoming obstacles to learning. The best estimates of the prevalence of mental health issues are that clinically significant depression and anxiety increased two-fold globally from prior to the pandemic to today, affecting 23.8% and 19% (respectively) of children and adolescents currently [13]. Thus, there is an acute need for secondary schools to effectively support adolescents' psychosocial adjustment and mental health needs today. Given gratitude's many advantages for students and schools [28], gratitude practices in school settings could provide easy, cost-effective ways to address this mental health crisis.

However, various studies indicate that among adolescents and young adults, females tend to be more grateful than males [29–34]. Such evidence indicates the need to better understand how gender can be considered in making gratitude interventions more effective for in-school adolescents. This is another goal of the current research—to examine if the individual components of the intervention examined by Bono et al. [22] have different effects on male vs. female students, because their study did not examine such effects.

### 1.4. Perceived Stress as a Mechanism for Gratitude's Mental Health and Well-Being Effects

One question that needs to be investigated regarding depression and anxiety in youth is: how can they be prevented? A risk factor that has been continuously linked to depression, anxiety, and SWB is perceived stress [35–37]. On the other hand, gratitude is an important protective factor that is linked to lower depression, lower anxiety, and higher SWB. Based on prior research, both gratitude and perceived stress contribute to anxiety, depression, and SWB [38–40]. However, these were findings from research with adults. What role do both gratitude and perceived stress play together in protecting adolescents from anxiety and depression and in supporting their SWB?

Research suggests that perceived stress may be a mechanism for how gratitude achieves well-being and mental health effects [30,33]. Indeed, evidence suggests that one reason gratitude benefits health is because it promotes a sense of coherence [41,42], which is considered a personal resource that includes confidence in oneself, environmental support, and the future [43], and guides one's reactions to stressful situations [44]. Thus, this study examines if trait gratitude's effects on stress appraisals help explain its relationships with mental health and well-being. However, evidence indicates that females tend to exhibit greater depression [45] and anxiety [46], but also greater SWB [47] than males during adolescence. Therefore, gender effects on perceived stress, depression, anxiety, and SWB should also be examined. However, one last question to address is: what has limited school gratitude interventions prior to Bono et al. [22] from impacting mental health outcomes?

### 1.5. The Need for Authentic Interpersonal Gratitude Practices in Intervention

It is important for schools to create "identity safe" spaces where different social identities are welcomed as assets to combat belonging uncertainty in adolescents [48]. Interventions that incorporate social context effectively are developmentally advantageous because they make the target behavior self-relevant to adolescents so that they feel meaningfully connected to something valuable and beyond themselves [49]. However, letting students and teachers thank each other in an identity-safe manner (i.e., authentically) is a quality that has been neglected in the design of school gratitude interventions. This quality may have helped the intervention examined by Bono et al. [22] be more supportive of adolescents' mental health and well-being, as the GiveThx app was designed with input from high school students to support psychological safety.

Therefore, to further help evaluate the effectiveness of the intervention examined by Bono et al., we explore whether students and teachers expressed psychological safety through the thanks messages they exchanged with each other during the intervention. This was carried out using informal qualitative analysis (i.e., content analysis of whether thanks messages showed any feelings of psychological safety in thankers) to evaluate the effectiveness of the intervention in achieving an identity-safe climate in the classroom.

### 1.6. Current Research

The current research first examines the effectiveness of the individual components in the intervention examined by Bono et al. [22]. Then, it examines changes in trait gratitude as a protective factor and changes in perceived stress as a mediating risk factor for anxiety, depression, and SWB among the high school adolescents who participated in the intervention. The gratitude intervention consisted of two components. One component was a psychoeducational curriculum named Thanks!: A Strengths-Based Curriculum for Teens and Tweens that teaches the science of gratitude and provides various practices (i.e., acknowledging each other's signature strengths, journaling, thank you letters to benefactors, and effective ways to express thanks to others) [50]. The other was a social media-like web-app called GiveThx, which allows students and teachers to exchange thanks in class, but privately, to avoid the competitive or harmful qualities common in social media [51]. GiveThx was designed to be inclusive of diverse social identities. Thus, we also searched for thanks messages exchanged between students and teachers in the intervention that demonstrated themes of feeling valued or accepted by others or of feeling belonging or trust with others to illustrate the climate of psychological safety created in the classes by the intervention [48].

Therefore, our first goal was to examine if (A) a psychoeducational approach and an approach of sharing interpersonal thanks with others each help male and female adolescents to different degrees and if (B) the use of both approaches together helps male and female adolescents more than either approach. We hypothesized that the findings from Bono et al. [22] would be supported—that is, that an intervention approach with both components combined (i.e., the full intervention) would impact all outcomes better than either intervention approach alone or a control group for all students. Given the reliable gender differences in gratitude found in the literature, we also hypothesized that the full intervention would impact females' growth in trait gratitude more than males and that the curriculum-only condition (i.e., the psychoeducational component) would generally impact outcomes for girls more than for boys.

Our second goal was to better explain the results of the intervention. We did this by examining if trait gratitude in the sample affects these outcomes via effects on perceived stress, if gender affects putative mediator (perceived stress) and the outcomes (depression, anxiety, and SWB), and if thanks messages reflect psychological safety in the intervention participants. We hypothesized that perceived stress would mediate the relationships of trait gratitude with depression, anxiety, and SWB from baseline to six weeks. Furthermore, we expected gender effects on perceived stress and the outcomes; see Figure 1 (below). However, we only hypothesize that females will exhibit greater increases in SWB than

males, given that females may be more receptive to practicing gratitude than males. Finally, given the emphasis of this intervention on interpersonal gratitude, we wanted to explore themes of trust and authentic disclosures in the thanks messages shared by students and teachers to evaluate if they felt psychological safety with the intervention activity, because we hypothesized that such a quality of experience in the intervention (i.e., feeling valued and more intimately connected with each other) makes thanking more meaningful and, therefore, beneficial to mental health and well-being.

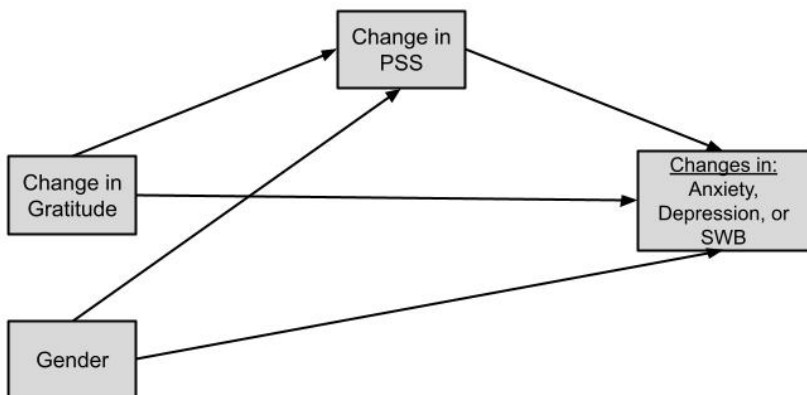

**Figure 1.** Theoretical model of the relationships between gratitude change and changes in anxiety, depression, or subjective well-being as mediated by perceived stress with a covariate of gender.

## 2. Materials and Methods

The research used a pretest–posttest quasi-experimental waitlist design in which 21 classrooms were randomly assigned to 1 of 4 conditions. Two conditions involved each of the two intervention components separately (Condition 1: Thanks! Curriculum, Condition 2: GiveThx web-app), a third condition involved both components together, and a fourth condition involved control classes that were either waitlisted or that remained pure controls (i.e., did not receive a change in their curriculum nor did they use the GiveThx app). See Table 1 for a basic description of the intervention. The number of students in each classroom ranged from 2–30, with the average classroom size being 15.52 (SD = 8.36) students.

**Table 1.** Basic description of school gratitude intervention evaluated in this study.

| Intervention Condition | Components | Dosage |
| --- | --- | --- |
| Psychoeducational curriculum | Included readings, slide decks, reflection activities, strengths inventories, and interpersonal gratitude tasks | 1 40-min lesson per week for 4 weeks |
| GiveThx app | Digital reflections, interpersonal gratitude messages, and strengths inventories | 5 min, twice per week for 6 weeks |
| Full intervention | Combined the curriculum and app components | Curriculum: 1 40-min lesson per week for 4 weeks App: 5 min, twice per week for 6 weeks |

### 2.1. Participants

The sample consisted of 326 students from 2 urban high schools (55% female and 45% male). The participants identified as 85% Hispanic, 10% Asian, 3% Black, 1% White, and 1% Other. The mean age of the sample was 14.72 years old (SD = 1.36) and the ages of the participants ranged from 13 to 18 years. Classes involved in the study were taught by 7 different teachers, and the following number of students were found across the 4 conditions: intervention involving just the curriculum (N = 16), intervention involving just the app (N = 46), the full intervention with both components (N = 120), and the control condition (N = 144). Not included in the sample were 176 students who did not complete the posttest survey. Despite efforts to obtain comparable sample sizes across the conditions, one teacher with three curriculum-only and two app-only classes had to switch classes to the control condition, which resulted in smaller sample sizes for those conditions.

*2.2. Procedure*

The study was approved by the authors' IRB board for meeting APA ethics standards. Parental consent was obtained from students who were minors, and then informed consent was obtained from all students at the start of the fall 2017 semester. Additionally, before intervention, teachers practiced using the app with fictitious characters and they participated in a workshop about the curriculum lessons and research protocol, where they were taught to prepare and deliver lessons and to administer the intervention evaluation surveys. Each teacher was compensated for their time with a $300 gift card. The study was conducted over the course of six weeks during that semester. In each condition, teachers administered a survey to participants electronically at week 1 (T1) and at week 6 (T2) of the study. Only students for whom consent was obtained completed surveys. However, all students participated in the same condition activities as a class. Finally, the administrator who co-created the app served as a facilitator to ensure that teachers implemented interventions with fidelity and adhered to the data collection schedule.

*2.3. Measures*

Student surveys contained a variety of variables, including demographic information, gratitude, common depression symptoms, common symptoms of general/social anxiety, positive and negative affect, life satisfaction, perceived stress, and other variables not included in this study.

To measure trait gratitude, the Gratitude Questionnaire 6 [52] was modified slightly in wording and scaling to tailor it for use with adolescents. Gratitude is defined as the tendency to be thankful and appreciative of benefits and benefactors in one's life. Participants rated how much the 6 items describe them using a 6-point scale (1 = "Strongly Disagree" to 6 = "Strongly Agree"). The scale had good internal consistency at T1 and T2 ($\alpha$'s = 0.84 and 0.82), with higher scores indicating more trait gratitude.

To measure depression, six items from the Center for Epidemiological Studies Depression Scale for Children [53] were used. Participants rated how often they felt common symptoms (i.e., down and unhappy, lonely, not as good as peers, too tired to do things, unable to be happy, poorer sleep) in the last 2 weeks, using a 5-point scale (0 = "Not at All" to 4 = "Always"). The scale had good internal consistency at T1 and T2 ($\alpha$'s = 0.75 and 0.79), with higher scores indicating more depression symptoms.

Common general and social anxiety symptoms were measured using six items from the Spence Children's Anxiety Scale [54]. Participants used a 5-point scale (0 = "Not at all" to 4 = "Always") to rate how much they worried about things, what other people thought of them, or about something bad happening, or how much they felt shaky when they had problems, afraid of making a fool of themselves in front of others, or afraid to talk in front of the class. The scale had good internal consistency at T1 and T2 ($\alpha$'s = 0.79 and 0.84), with higher scores indicating more anxiety symptoms.

Perceived stress involves thoughts and feelings of whether life circumstances at any given time are manageable or overwhelming. The Perceived Stress Scale 4 was used to measure such stress appraisals [55]. Participants rated the 4 items on a 5-point scale (0 = "Not at All" to 4 = "Always"). Internal consistencies were low at T1 and T2 ($\alpha$'s = 0.51 and 0.62). The few items and reverse-scoring for half of them may have been a factor with our sample of youth. However, test–retest among the control group was high (r = 0.51, $p < 0.001$), suggesting acceptable reliability. Higher scores indicated greater stress.

To assess subjective well-being (SWB), a composite measure was created using two items from the Students' Life Satisfaction Scale [56] and 19 items from the Positive and Negative Affect Scale for Children [57], which included 11 PA items and 8 NA items. Participants used a 6-point scale to rate how much their life is going well or that they had a different life lately (1 = "Never/Almost Never" to 4 = "Always/Almost Always") and how much they experienced different positive and negative emotions—such as happy, proud, mad, and irritable—in the last month (1 = "Never/Almost Never" to 4 = "Always/Almost Always"). To create SWB, the negative life satisfaction item and the NA items were reverse-

scored and averaged together with the positive life satisfaction and PA items into a single composite variable. This adheres to a commonly used method [58,59] and resulted in a scale with good internal consistency at T1 and T2 ($\alpha$'s = 0.88 and 0.89), with higher scores indicating more SWB.

*2.4. Analyses*

Data were analyzed using SPSS 28. Difference scores for each variable (gratitude, depression, anxiety, perceived stress, SWB) were calculated for each respondent by subtracting T1 scores from T2 scores. Thus, positive change scores reflected an increase in a construct, whereas negative change scores reflected a decrease in a construct.

Given that the hierarchical structure of the data (i.e., students (level-1) nested within 21 classrooms (level-2)) violates the assumption of independence in a traditional ordinary least squares framework, the planned analyses for the first study goal were to use multilevel modeling (MLM) [60] within SPSS mixed models. MLM assumes that level-1 observations are dependent within each cluster (classroom) and that fixed and random effects are estimated simultaneously within the same model. That is, the dependence was considered "nuisance" variability to be disaggregated from the fixed parameters (i.e., mean differences between groups on the outcome variables). MLM was planned using REML estimation, given that models would not be compared.

Four models were run (for each outcome of change in gratitude, depression, anxiety, and SWB). Classroom was modeled as a random effect. Fixed effects included the covariate of age, the main effect of intervention condition (four conditions, described above), the main effect for gender, and the experimental condition X gender interaction (to determine whether intervention effects were modified by gender).

For the second study goal, we first used bivariate correlations to assess the basic relationships between all the variables. We then tested our hypotheses for mediation and gender effects using the PROCESS macro for SPSS [61]. Lastly, we explored the thanking messages that teachers and students exchanged during the intervention for themes related to psychological or identity safety (i.e., feeling trust or belonging, intimate disclosures, appreciating caring and commitment from others). This qualitative analysis was informal, serving to evaluate if students and teachers exhibited psychologically safe thanking in the classrooms while using the GiveThx app.

## 3. Results

### 3.1. Results of MLMs

Descriptive statistics of the dependent variables at baseline (T1) and posttest (T2)—gratitude, depression, anxiety, and SWB—are presented in Table 2 separately by gender across the three experimental conditions and the control condition. The means that trait gratitude scores across all conditions are quite high for males and female students (six is the maximum rating), indicating ceiling effects. Table 3 presents the descriptive statistics of change scores in these dependent measures across all four conditions. Table 4 presents the effects of the gratitude intervention on trait gratitude, depression, anxiety, and SWB. The models for SWB and anxiety would not converge due to null variance accounted for by classrooms in these models. Given this, and the relatively low number of observations for some classrooms, a traditional ordinary least squares framework was utilized for these two models.

There were intervention effects for gratitude, anxiety, and SWB. However, none of these effects were modified by gender ($ps \geq 0.162$), suggesting that there were not gender differences across the different intervention approaches for any of these outcomes. More specifically, respondents in the Curriculum + app condition had significantly greater SWB ($p < 0.001$) and gratitude ($p = 0.010$), and significantly lower anxiety ($p = 0.004$) compared to respondents in the Control condition. There was an additional effect for SWB wherein those who were in the Curriculum + app condition had significantly greater SWB compared to those in the App-only condition ($p = 0.011$). There were no other significant pairwise comparisons.

**Table 2.** Descriptive statistics of dependent variables at T1 and T2, by condition and gender.

| Condition | Gender | Time | Gratitude | Depression | Anxiety | SWB |
|---|---|---|---|---|---|---|
| Control | Male | T1 | 4.86 (1.00) | 1.82 (1.10) | 1.61 (0.62) | 4.29 (0.69) |
| | | T2 | 4.41 (0.97) | 1.80 (0.93) | 1.85 (0.70) | 4.20 (0.71) |
| | Female | T1 | 4.87 (0.99) | 2.30 (1.10) | 2.4 (0.84) | 3.81 (0.70) |
| | | T2 | 4.68 (1.05) | 2.06 (0.81) | 2.1 (1.04) | 3.80 (0.77) |
| Curriculum only | Male | T1 | 4.95 (0.80) | 1.51 (1.34) | 1.60 (1.12) | 4.38 (0.57) |
| | | T2 | 4.82 (0.80) | 1.69 (0.98) | 1.88 (1.00) | 4.35 (0.63) |
| | Female | T1 | 4.57 (0.88) | 1.97 (1.01) | 1.91 (0.78) | 3.90 (6.1) |
| | | T2 | 4.57 (1.02) | 1.87 (0.86) | 2.13 (0.83) | 3.81 (0.67) |
| App-only | Male | T1 | 4.87 (1.07) | 1.95 (1.40) | 1.85 (0.95) | 4.00 (0.84) |
| | | T2 | 4.68 (0.73) | 1.60 (0.99) | 1.61 (1.10) | 4.16 (0.78) |
| | Female | T1 | 4.97 (0.76) | 1.64 (0.95) | 2.13 (0.84) | 4.14 (0.78) |
| | | T2 | 5.14 (0.73) | 1.78 (0.95) | 2.13 (0.87) | 4.07 (0.79) |
| Curriculum + app | Male | T1 | 5.01 (0.73) | 1.70 (1.12) | 1.72 (0.95) | 4.22 (0.63) |
| | | T2 | 4.92 (0.79) | 1.50 (0.87) | 1.67 (0.81) | 4.37 (0.64) |
| | Female | T1 | 4.87 (0.75) | 1.94 (1.00) | 1.82 (0.78) | 3.90 (0.50) |
| | | T2 | 4.88 (0.77) | 1.60 (0.87) | 1.79 (0.88) | 4.05 (0.57) |

Note: SWB = subjective well-being.

**Table 3.** Mean differences from baseline to six weeks across conditions.

| | Trait Gratitude | | | Depression | | | Anxiety | | | SWB | | |
|---|---|---|---|---|---|---|---|---|---|---|---|---|
| | *n* | *M* | *SD* | *n* | *M* | *SD* | *n* | *M* | *SD* | *n* | *M* | *SD* |
| Control | 144 | −0.21 | 0.56 | 141 | −0.13 | 1.25 | 143 | 0.19 | 0.91 | 144 | −0.03 | 0.40 |
| Curriculum | 16 | 0.20 | 0.44 | 16 | −0.08 | 1.07 | 16 | −0.17 | 0.93 | 16 | 0.03 | 0.31 |
| App | 46 | 0.01 | 0.69 | 46 | −0.09 | 1.38 | 46 | −0.11 | 0.71 | 46 | 0.03 | 0.49 |
| Combination | 120 | 0.09 | 0.61 | 119 | −0.30 | 1.19 | 120 | −0.15 | 0.69 | 120 | 0.22 | 0.40 |

Note: conditions include a control and three types of gratitude interventions (Thanks Curriculum alone, GiveThx app alone, and a combination of the app and curriculum).

**Table 4.** Impacts of gratitude interventions on gratitude, mental health, and well-being.

| Fixed Effects | Trait Gratitude | Depression | Anxiety | SWB |
|---|---|---|---|---|
| Age | $F(1, 14.43) = 0.10$, $p = 0.755$ | $F(1, 26.56) = 2.10$, $p = 0.159$ | $F(1, 315) = 4.10$ *, $p = 0.044$ | $F(1, 316) = 0.74$, $p = 0.391$ |
| Gender | $F(1, 303.48) = 5.71$ **, $p = 0.017$ | $F(1, 304.29) = 0.18$, $p = 0.669$ | $F(1, 315) = 0.01$, $p = 0.943$ | $F(1, 316) = 0.03$, $p = 0.871$ |
| Condition | $F(3, 6.83) = 6.32$ **, $p = 0.022$ | $F(3, 11.88) = 0.37$, $p = 0.776$ | $F(3, 315) = 3.53$ **, $p = 0.015$ | $F(3, 316) = 7.01$ ***, $p < 0.001$ |
| Condition x Gender | $F(3, 306.14) = 0.39$, $p = 0.764$ | $F(3, 306.30) = 1.73$, $p = 0.162$ | $F(3, 315) = 1.04$, $p = 0.373$ | $F(3, 316) = 1.34$, $p = 0.262$ |

Note: the models for SWB and anxiety used OLS; the models for depression and gratitude used MLM. Conditions include three types of intervention classes (GiveThx app alone, Thanks Curriculum alone, and App + Curriculum) and control classes. * $p < 0.05$, ** $p \leq 0.02$, *** $p < 0.001$.

### 3.2. Results of Mediation Analyses

A bivariate correlation analysis was first conducted to investigate the relationships between all the variables in question. Gratitude change was positively correlated with being female ($r = 0.15$, $p = 0.009$) and with SWB change ($r = 0.37$, $p < 0.001$) and negatively correlated with changes in anxiety ($r = −0.21$, $p < 0.001$) and depression ($r = −0.18$, $p = 0.001$). Receiving any gratitude intervention was negatively correlated with changes in perceived stress ($r = −0.16$, $p = 0.005$) and anxiety ($r = −0.20$, $p < 0.001$) and positively correlated with changes in trait gratitude ($r = 0.24$, $p < 0.001$) and SWB ($r = 0.22$, $p < 0.001$).

The PROCESS macro for SPSS was used to test three mediation models with the antecedent variable of gratitude change, the mediator of perceived stress change, the

outcome variables of anxiety change, depression change, or SWB change, and the covariate of gender. The statistical significance of the indirect effect was tested using bootstrapping procedures. In each of these mediation models, there was a gender difference in perceived stress change (b = −0.65, $p$ = 0.04), with males having a greater change in perceived stress. The path representing the relationship between gratitude change and perceived stress change in each of the three mediation models was statistically significant, b = −1.17, $p < 0.001$.

Anxiety. The path representing the relationship between gratitude change and anxiety change was statistically significant, b = −0.24, $p < 0.001$. The path representing the relationship between perceived stress change and anxiety change was also statistically significant, b = 0.08, $p < 0.001$. There was a significant indirect effect of gratitude change on anxiety change through perceived stress change, b = −0.10, 95% CI [−0.16, −0.05]. The direct path from gratitude change to anxiety change approached significance (b = −0.14, $p$ = 0.06), indicating a trend for perceived stress change partially mediating the relationship between gratitude change and anxiety change; see Figure 2.

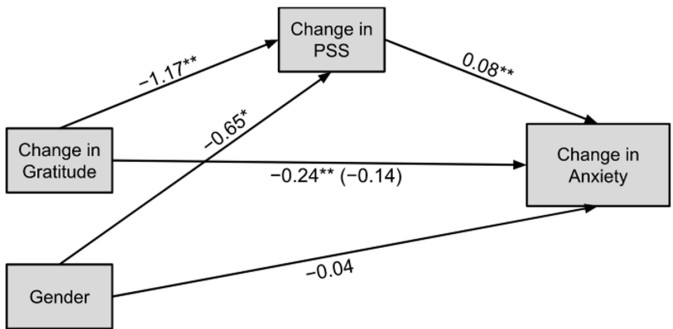

**Figure 2.** Coefficients for the relationship between change in gratitude and change in anxiety as mediated by change in perceived stress with a covariate of gender. The coefficient between change in gratitude and change in anxiety, controlling for change in perceived stress, is in parentheses. * $p < 0.05$, ** $p < 0.001$.

Depression. The path representing the relationship between gratitude change and depression change was statistically significant, b = −0.34, $p < 0.001$. The path representing the relationship between perceived stress change and depression change was also statistically significant, b = 0.05, $p$ = 0.04. For the indirect effect of gratitude change on depression change through change in perceived stress, there was a trend towards significance, b = −0.06, 95% CI [−0.14, 0.01]. Because the direct path from gratitude change to depression change remained significant (b = −0.27, $p$ = 0.02), this shows that change in perceived stress partially mediates the relationship between gratitude change and depression change; see Figure 3.

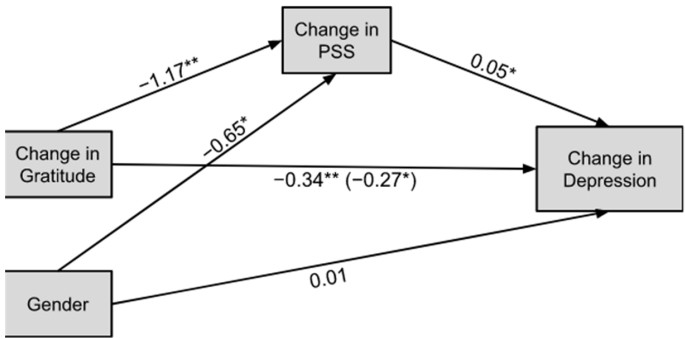

**Figure 3.** Coefficients for the relationship between change in gratitude and change in depression as mediated by change in perceived stress with a covariate of gender. The coefficient between change in gratitude and change in depression, controlling for change in perceived stress, is in parentheses. * $p < 0.05$, ** $p < 0.001$.

SWB. There was a significant gender difference in SWB change (b = 0.10, *p* = 0.02), with females exhibiting greater change in SWB. The path representing the relationship between gratitude change and SWB change was statistically significant, b = −0.24, *p* < 0.001. The path representing the relationship between perceived stress change and SWB change was also statistically significant, b = 0.05, *p* < 0.001. There was a significant indirect effect of gratitude change on SWB change through change in perceived stress, b = −0.05, 95% CI [−0.09, −0.02]. Because the direct path from gratitude change to SWB change remained significant (b = −0.19, *p* < 0.001), these results suggest that perceived stress change partially mediates the relationship between gratitude change and SWB change; see Figure 4.

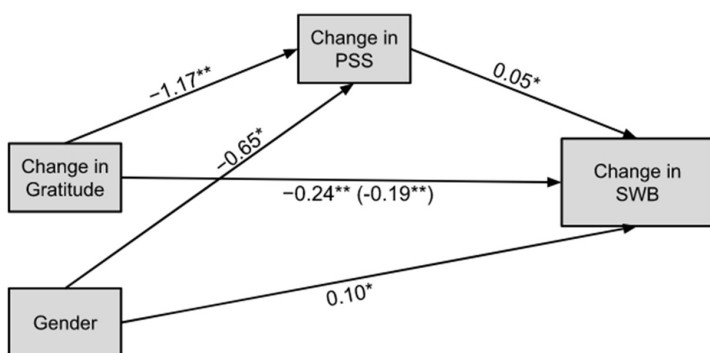

**Figure 4.** Coefficients for the relationship between change in gratitude and change in subjective well-being as mediated by change in perceived stress with a covariate of gender. The coefficient between change in gratitude and change in subjective well-being, controlling for change in perceived stress, is in parentheses. * *p* < 0.05, ** *p* < 0.001.

*3.3. Informal Qualitative Analysis of Thanks Messages*

Finally, examples of psychologically safe thanks messages sent by students and teachers during the intervention are provided in Table 5. For this informal qualitative analysis, we simply identified thanks messages that demonstrated themes of feeling valued or accepted, feeling trust or belonging, or appreciating social or emotional support from others. No reliability analyses were carried out, and the first author simply analyzed some of the thanks messages exchanged during the intervention that had content related to these themes. They illustrate the kind of genuine thanking that can occur when interventions create an environment that is safe or inclusive of everyone's social identity [41].

**Table 5.** Examples of psychologically safe thanks shared by students and teachers in class during the gratitude intervention.

| Participant | Thanks Notes Sent by Students and Teachers to Each Other |
|---|---|
| | *Students Thanking Peers* |
| Student 1 | Hi Sandy! I want to appreciate how you're always so friendly and make me feel welcomed whenever I see you in class or around school. |
| Student 2 | Hey, Marcus thanks for being a huge help and just always making me laugh even when I'm hella sad. Your word is great. |
| Student 3 | Thank you for laughing at my jokes and spending time with me in advisory :) |
| Student 4 | Thank you for always staying positive through the good and bad. You're a huge motivation for me, you help me to keep going with everything by being by my side. I will always and forever be here for you. Don't forget. |
| Student 5 | Thank you for helping me out all the time even though you aren't working on the same project as me anymore. |
| Student 6 | I want to thank you for being a good friend and also for being a great and supportive partner in English. |
| | *Students Thanking Teachers* |
| Student 7 | Thanks for sharing with me your ideas on how I could improve on my portrait! |
| Student 8 | Hi Mr. Keating. Thanks for helping me with shading. I don't think you are mean because you sent me back a couple times to work on it. I understand that to get better results practice is needed. |
| Student 9 | Thanks for making this period as fun and as enjoyable as it could be. I know we were just cutting circles, but it was pretty fun! |
| | *Teachers Thanking Students* |
| Teacher 1 | I saw you really leading your group. Thank you for making sure everything was completed and cleaned up in class today. |
| Teacher 2 | Hi Isabela, I really appreciate every time that you participate in class. I know that you consider yourself to be shy. So every time that you participate it shows me that you are becoming more comfortable in my class! Thank you! |
| Teacher 3 | Hi Jeremy, I appreciate how much of an advocate you are. Through emails and face to face questions, you are a strong, hard-working student. You help me become a better teacher. |
| Teacher 4 | I heard you asking your peers if they needed helping taking pictures and I appreciate how you are willing to help others |
| Teacher 5 | Jasmine, I want to appreciate you for being so engaged in class and always asking interesting questions that help push the class's thinking. |

## 4. Discussion

Results of the first analyses replicated the main findings from the previous research [22], showing that the full gratitude intervention impacted gratitude, anxiety symptoms, and well-being six weeks later, compared to a control group (see Table 3 above). Further, the use of more appropriate analytic methods (i.e., MLM) with these data that were nested within classrooms did not yield different findings from the prior study. However, unlike the prior study, this study provides statistical evidence that combining a psychoeducational approach (use of a curriculum) with a modality of interpersonal practice (use of an app for thanking peers and teachers) impacted mental health and well-being more than using neither and that the full intervention also impacted well-being more than the app-only condition. This coincides with evidence that meaningfully thanking benefactors is particularly potent for affecting adults' well-being [62–64]. Importantly, the findings suggest that teaching the science and reasoning behind gratitude's benefits makes the practice of thanking others more meaningful to students.

Therefore, gratitude interventions conducted in schools should not only provide a way for students and teachers to give thanks naturally and authentically during the school day, but they should also help participants understand why thanking others for kindness is important for improving our social relationships' well-being. While various forms of gratitude may be beneficial, our results suggest that sharing genuine thanks with benefactors in school settings—such as friends, teachers, or staff—better supports students' mental health and well-being. Given recent global trends in adolescents' anxiety and depression [14], effective gratitude interventions in schools could provide an easy, scalable resource to address the mental health challenges among youth that continue to be exacerbated by the COVID pandemic.

It is noteworthy that while research has found school gratitude interventions can be effective in promoting trait gratitude and well-being when they include an interpersonal component [21,22], as far as we know, effects of school gratitude interventions on mental health outcomes such as anxiety have not yet been documented. Thus, our findings corroborated Bono et al.'s intervention approach [22] as a viable way to promote well-being and mental health in high school students.

As Table 2 shows, the control condition exhibited reductions in trait gratitude by the end of the semester. Thus, comparison against this pattern in the control group is what yielded significant effects on trait gratitude for the full intervention (curriculum and app) relative to the control condition. It seems likely that ceiling effects may be a reason for the limited increases in trait gratitude that were observed for the full intervention. However, a similar pattern occurred with anxiety and SWB, which worsened in the control group but not the full intervention group. Such downward cyclical trends in well-being indicators are commonly found in research on school interventions [65] and may be partly due to increased academic demands toward the end of the semester. Therefore, the full intervention appeared to counter this trend in terms of impacting anxiety and SWB relative to the control group as well.

Though Bono et al. [22] used OLS analyses in finding that the full intervention (i.e., using both the app and curriculum) impacted depression for boys only, compared to the control group, the MLM analyses here did not support this hypothesis. We also did not find support for our hypotheses that females would benefit more from the curriculum alone than males.

Our second set of analyses sought to better explain how trait gratitude is benefiting the mental health and well-being of the adolescents in the intervention by testing whether changes in trait gratitude achieved their effects on the outcomes via changes in perceived stress and investigating the relationships of gender with perceived stress and the outcome variables. We predicted that perceived stress would be a significant mediating variable for the relationships of trait gratitude with depression and anxiety. This hypothesis was supported, as the relationship between change in gratitude and change in depression was partially mediated by change in perceived stress, and the relationship between change in gratitude and change in anxiety was totally mediated by change in perceived stress.

We also predicted that perceived stress would be a significant mediating variable for the relationship between trait gratitude and SWB. This hypothesis was also supported. The relationship between change in gratitude and change in SWB was partially mediated by change in perceived stress. Secondly, we predicted that there would be gender differences in perceived stress, depression, anxiety, and SWB. This hypothesis was partially supported, as a gender difference was found in perceived stress, with males exhibiting greater change in perceived stress than females, and a gender difference was found in SWB, with females exhibiting greater change in SWB than males.

Finally, we also wanted to evaluate if the intervention provided a psychologically safe space [48] for students and teachers by exploring whether thanks notes exchanged by students and teachers during the intervention illustrated intimate feelings of trust and social belonging (see Table 5 above). These thanks notes illustrate how intimate disclosure and authentic thanking can make in-person interactions more meaningful, create a sense of caring and connection in the classroom by promoting stronger social relationships between students and peers and between students and teachers. Though the breadth and depth of psychological safety inherent in our intervention participants' thanking behavior was not examined directly and systematically with formal qualitative analysis of all the thanks messages exchanged during the intervention, it was evident that this quality was achieved by the intervention. Identity safety was explicitly designed into the GiveThx app, and this represents a unique feature of the intervention evaluated here and previously [22] that has been largely neglected in other school gratitude interventions. The qualitative results suggest that an identity-safe climate in the intervention classes may have contributed to the mental health effects observed in our adolescent sample. Therefore, researchers, educators, and practitioners should seek to achieve identity safety in classroom interventions promoting gratitude to improve their effectiveness.

Another important quality for positive psychology interventions in general that characterized the intervention evaluated here is person–activity fit; that is, providing various activities that are novel and enjoyable, letting participants select practices that are intrinsically motivating, and persuading participants to exert effort on the activities [66]. The use of both curriculum and app-thanking components in the intervention, as well as the variety of gratitude practices enabled by these components, support this quality and may be another reason for the intervention's effectiveness. Interventions that are unresponsive to the needs and daily context of individuals' lives can provoke reactance in adolescents [67,68] and prevent them from self-disclosing. Indeed, research indicates that self-disclosure—or revealing one's feelings, thoughts, and emotions—is conducive to psychological and relational well-being [69]. Such self-disclosure, as illustrated in the thanks messages shared by students and teachers in the intervention, is important for building greater belonging and interpersonal well-being in school settings.

One limitation of this study is the short intervention duration of six weeks. To grow gratitude habits—and produce stronger mental health and well-being effects—gratitude practices should be maintained for longer periods of time and be supported by teachers and other staff at school throughout the school year as part of the school culture. Ideally, schools should implement a variety of gratitude practices sustainably with students (i.e., in ways that are responsive to their social lives and preferences and that also support their academic success) and sustainably as part of their culture (i.e., with buy-in from all staff and administration and adherence to the school's values). Another limitation of this study is that the quasi-experimental groups were not balanced, as comparable numbers were not obtained across the four conditions. Thus, results for the curriculum-only condition could be different with a larger sample.

Nonetheless, the current study found that combining a psychoeducational approach and providing an identity-safe modality to practice gratitude personally and interpersonally appear to be two important emphases for interventions to be effective. Therefore, future intervention efforts in this area will benefit from establishing strong coordination between educators, researchers, and mental health and counseling staff; and schools should

be creative in implementing a variety of engaging practices, with a great number of opportunities for adults and youth to naturally practice gratitude and kindness, as well as reflect on these positive behaviors, regularly at school.

## 5. Conclusions

The current research showed that engaging high school students in personal and interpersonal practices of gratitude is effective for supporting their mental health and well-being, and that one reason gratitude habits are beneficial is because they help individuals' broad orientations for coping with daily challenges (i.e., perceived stress). The intervention helped both males and females in terms of their social and general anxiety symptoms and their emotional well-being and SWB. Our findings suggest that school gratitude interventions should provide students and teachers various ways to reflect gratefully and to give thanks to each other naturally and authentically during the school day. For example, such opportunities can be provided during collaborative assignments to emphasize the strategies and strengths of students that helped produce success for teams. Advisory or homeroom classes can use gratitude practices to start off the day positively and with optimism to help students view school as a place where everyone is valued. Discussions could become more meaningful and engaging from the greater classroom cohesion.

School counselors, psychologists, teachers, and staff can each help students appreciate the resources and support provided by administrators for new programs, by support staff for special events, or by teachers for using fun or engaging teaching strategies. When adults and students place time and effort into helping each other at school and then acknowledge each other during such moments, this creates a more caring community where students can feel inspired to contribute their best efforts to overcome challenges and strive for success.

A major challenge of school gratitude interventions is persuading students and teachers to thank each other genuinely and autonomously so that kindness is reinforced whenever it occurs, whether in classrooms or in the hallways or during or after school. Importantly, providing identity-safe ways to practice gratitude helps adolescents to self-disclose about distressing issues that otherwise could elicit disruptive conduct or impede attention and learning. Interventions that are responsive to the daily context and different identities of students and teachers enable both parties to develop adaptive meanings through their social interactions, and this not only helps them flourish, but it increases equity in schools by making the intervention practices more accessible and useful for everyone [68].

Rates of adolescents' anxiety and depression were already rising in societies across the globe, and the COVID pandemic only exacerbated matters in the last few years, especially for minorities. Students need supportive communities to deal with unprecedented challenges and uncertainty in our world more than ever. Effective gratitude interventions in schools are not only cost-effective, but they offer an easy, scalable way to address the mental health challenges of the students. Many schools are struggling to provide sufficient levels of such support to students [6]. The current study provides recommendations for how schools can use gratitude practices more effectively to complement and maybe even improve their social emotional learning programs. Our results suggest that doing so will better support the mental health and well-being of students. However, it is entirely possible that more creative efforts sustained school-wide as part of school culture throughout the year could produce even stronger benefits, not just to mental health and well-being, but to other outcomes that are valuable for schools, such as academic achievement and school climate.

**Author Contributions:** Conceptualization, G.B.; methodology, G.B., E.L.M. and T.D.; software, E.L.M. and T.D.; validation, T.D.; formal analysis, E.L.M. and T.D.; investigation, G.B.; resources, G.B.; data curation, G.B.; writing—original draft preparation, G.B., E.L.M. and T.D.; writing—review and editing, G.B., E.L.M. and T.D.; visualization, T.D. and E.L.M.; supervision, G.B.; project administration, G.B. All authors have read and agreed to the published version of the manuscript.

**Funding:** The first author G.B. conducted the original intervention research with funding by the JOHN TEMPLETON FOUNDATION, grant #55309.

**Institutional Review Board Statement:** This study was approved by the Institutional Review Board for the Protection of Human Subject of California State University Dominguez Hills (#15-225) on 12 September 2017.

**Informed Consent Statement:** Written informed consent was obtained from the study participants and their guardians.

**Data Availability Statement:** Data available on request from the corresponding author G.B.

**Conflicts of Interest:** The authors declare no conflict of interest. The funders had no role in the design of the study; in the collection, analyses, or interpretation of data; in the writing of the manuscript; or in the decision to publish the results.

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
