# Peer review of "Gratitude and Adolescents’ Mental Health and Well-Being: Effects and Gender Differences for a Positive Social Media Intervention in High Schools"

_education, doi:10.3390/educsci13030320_

Round 1

Reviewer 1 Report

Very good article, well structured and well written.

I would only suggest two changes:

1st the table 1 give it the form of a table providing more information or reordering the information.

2º In the description of the participants determine the gender of the different origins of the participants.

Thank you very much.

Author Response

Thanks for the feedback Reviewer 1. Our revision has addressed your critiques. Table 1 is now properly formatted and the gender of the participants is clearly indicated in the participants section.

Reviewer 2 Report

This is an evaluative research, combining quantitative and qualitative methods. 

The abstract must give a clear answer: how do you evaluate? 

rows 13 - 24 need reformulation, insert here:

- the purpose of the research

- research questions (hypothesis)

- methodology (qualitative and quantitative!)

- variables and how were measured

- the qualitative analyses strategy

- what is the study design?

- the study sample

- results

- too many keywords, delete some (just 4-5)

Methodology: describe separately the qualitative and quantitative methods with explications.

2.4 change in `quantitative variables and measuring`+ insert here the dependent and independent variables of the study

rows 309 - 391

Tables 3, 4 need some explication! Insert some rows here!

row 392-397

Insert table 6

Insert the qualitative analyses here!

rows 398 - 504

Insert here the discussion of the qualitative analyses.

rows 505 - 538

Return to your statistically significant results and insert here some phases when you talk about the practical significance of your finding. 

Other suggestions: 

Insert the explanation of the hole abbreviation!

row 539

The backend program was removed and here is the and of the peer - review!

I wish you all the best!

Author Response

Thanks Reviewer 2 for the feedback. It was helpful in improving the manuscript in the revision. We have addressed all of the concerns of Reviewer 2.

I describe the revisions we made which address the ratings that indicate potential improvement of the manuscript first. We restructured the sections of the literature review, made the literature review more concise throughout, and made the distinctions between the intervention, the first examination of the intervention (Bono et al., 2020), and this current study crystal clear - so that our study's basis in theory and our study's contributions to the empirical work in this area of research is much clearer now.  We believe that these revisions to the literature review have greatly improved the flow of the manuscript and addressed other critiques in the ratings too. For instance, the relevance of the citations is clearer now, the adequacy of the references is clearer, and the arguments made in the manuscript are now much clearer too.

Second, we have included the hypotheses in the abstract, made the hypotheses clearer in the literature review, clarified the methods in the abstract and in the body of the manuscript, and made the purpose of the study (which is to share best practices in the field with educators and researchers) clearer and more explicit too. So, for example, how we evaluated the intervention quantitatively and qualitatively in the current study, the design of the current study compared to the previous Bono et al. study, and the analytic methods used in the current study are much more clearly described now. While we believe that this helped set up the presentation of results to be clearer, we also directly revised the results presentation so that descriptions of the study elements are more consistent with descriptions throughout the paper too. Lastly, we made the conclusions clearer by explicitly describing how our results support the purpose of the study (i.e., how the best practices we have identified help strengthen intervention and the importance of considering gratitude intervention as a valuable underused tool for improving SEL).

Finally, we have addressed all the critiques in your comments too. First, we reduced our key words to 5 and included all the information requested in the abstract. Second, we clarified the title of Table 3 so that it clearly describes the dependent variables as "Mean Differences from Baseline to Six Weeks Across Conditions", and we included stars to indicate levels of statistical significance of the findings in Table 4. We clarified how the qualitative analysis was conducted not just in section 3.3. - where those results are first presented - but throughout the manuscript too. Also, we included Table 5 (the qualitative results, which Reviewer 2 accidentally referred to as "Table 6"); we apologize for forgetting to include Table 5 in the first manuscript submission! This undoubtedly made our discussion of the qualitative results confusing before and believe that the revision has greatly improved this aspect of our study. Lastly, we included some sentences in the conclusion to better discuss the practical significance of our study's findings (how SEL can be improved by including gratitude interventions that use the best practices emphasized by our study). 

Round 2

Reviewer 2 Report

Dear Authors,

Bono et all (2020) must be nr. 1 in references! It is included in your Abstract also.

Insert the explication of the abbreviation PROCESS (like a footnote or page note). The readers of the abstract must understand what is this.   

Change TABLE 5 in Table 5.

Author Response

Thanks for your additional, helpful feedback, Reviewer 2!

We made Bono et al. (2020) number 1 in the references and renumbered the citations and references accordingly.

We also changed the title of Table 5 from “TABLE 5” to “Table 5”.

Regarding clarifying "PROCESS", PROCESS is a modeling tool or macro that is commonly used to run mediation analyses in multiple statistical software programs. "PROCESS" is the formal name, and it is not an acronym or abbreviation.  However, we recognize that this may be confusing in the abstract to a reader, so we have edited the language in the abstract by removing any mention of PROCESS and instead just saying: "Then, stress was examined as a mediator of these effects." The formal reference for the macro is documented in the methods section of the paper, and we believe that this change and our documentation on that in the methods provide sufficient clarity now. Thanks again!